# The Opportunities and Challenges of Biobased Packaging Solutions

**DOI:** 10.3390/polym17162217

**Published:** 2025-08-14

**Authors:** Ed de Jong, Ingrid Goumans, Roy (H. A.) Visser, Ángel Puente, Gert-Jan Gruter

**Affiliations:** 1Avantium, Zekeringstraat 29, 1014 BV Amsterdam, The Netherlands; ingrid.goumans@avantium.com (I.G.); roy.visser@avantium.com (R.V.); 2Nova-Institut GmbH, Leyboldstraße 16, 50354 Hürth, Germany; angel.puente@nova-institut.de; 3Van’t Hoff Institute of Molecular Sciences, University of Amsterdam, Science Park 904, 1098 XH Amsterdam, The Netherlands

**Keywords:** biobased plastics, sustainable polyesters, sustainable packaging, circular materials, plastic recycling, biodegradable plastics, PEF, PLA, PHA

## Abstract

The outlook for biobased plastics in packaging applications is increasingly promising, driven by a combination of environmental advantages, technological innovation, and shifting market dynamics. Derived from renewable biological resources, these materials offer compelling benefits over conventional fossil-based plastics. They can substantially reduce greenhouse gas emissions, are often recyclable or biodegradable, and, in some cases, require less energy to produce. These characteristics position biobased plastics as a key solution to urgent environmental challenges, particularly those related to climate change and resource scarcity. Biobased plastics also demonstrate remarkable versatility. Their applications range from high-performance barrier layers in multilayer packaging to thermoformed containers, textile fibers, and lightweight plastic bags. Notably, all major fossil-based packaging applications can be substituted with biobased alternatives. This adaptability enhances their commercial viability across diverse sectors, including food and beverage, pharmaceutical, cosmetics, agriculture, textiles, and consumer goods. Several factors are accelerating growth in this sector. These include the increasing urgency of climate action, the innovation potential of biobased materials, and expanding government support through funding and regulatory initiatives. At the same time, consumer demand is shifting toward sustainable products, and companies are aligning their strategies with environmental, social, and governance (ESG) goals—further boosting market momentum. However, significant challenges remain. High production costs, limited economies of scale, and the capital-intensive nature of scaling biobased processes present economic hurdles. The absence of harmonized policies and standards across regions, along with underdeveloped end-of-life infrastructure, impedes effective waste management and recycling. Additionally, consumer confusion around the disposal of biobased plastics—particularly those labeled as biodegradable or compostable—can lead to contamination in recycling streams. Overcoming these barriers will require a coordinated, multifaceted approach. Key actions include investing in infrastructure, advancing technological innovation, supporting research and development, and establishing clear, consistent regulatory frameworks. Public procurement policies, eco-labeling schemes, and incentives for low-carbon products can also play a pivotal role in accelerating adoption. With the right support mechanisms in place, biobased plastics have the potential to become a cornerstone of a sustainable, circular economy.

## 1. The Necessity to Defossilize the Materials Sector

In addition to decarbonizing the energy sector, it is critically important to defossilize the chemicals and materials industries to prevent fossil-derived carbon from ultimately entering the atmosphere [1]. Notably, around two-thirds of the total carbon footprint of plastics stems from the carbon embedded within the polymer itself. If global polymer production continues to grow at an annual rate of 3.5%, and energy-related emissions are reduced by 90% by 2050, the carbon footprint of plastics alone could reach three gigatonnes of CO_2_—potentially dominating the remaining carbon budget [2]. Therefore, defossilizing the chemicals and plastic sector is not just a complementary effort—it is a critical driver to lower carbon emissions.

The Renewable Carbon Initiative (RCI), a collaborative effort aimed at defossilizing the chemicals and materials industry, has developed policy recommendations that enable the transition to a fossil-free chemical industry. The integration of renewable carbon targets in regulations is one of the key measures that addresses the root of the climate problem by halting the inflow of fossil carbon and incorporating embedded carbon in chemicals and derived materials into EU policies. Renewable carbon refers to carbon sources that are continuously replenished on a human timescale and do not contribute to the net increase in carbon dioxide in the atmosphere when used. These sources include biomass (e.g., plants, agricultural residues, algae), CO_2_ captured from the atmosphere or industrial processes, and recycled carbon from existing materials (e.g., plastics). Comprehensive carbon management must become a core component of these policies, organizing carbon as a circular resource across all industrial sectors and facilitating the transition from fossil carbon to renewable energy and renewable carbon [3].

Effective carbon management in Europe should also support the transformation of biofuel plants into chemical suppliers. Converting existing chemical infrastructure from fossil to renewable carbon is essential to achieve the necessary volumes and create a significant impact. This transformation requires a massive expansion of renewable energies, such as solar and wind power, and the development of green hydrogen grids. Carbon capture and utilization (CCU) can play a complementary role, particularly as a means of storing and transporting renewable energy and contributing to the production of renewable materials [4].

To support market access for renewable carbon-based products, binding renewable carbon quotas should be established across relevant product categories. Regulatory frameworks and public discourse should also positively recognize the role of atmospheric carbon utilization through CCU. However, this must not overshadow the critical importance and scalability of biobased carbon pathways, which are essential for a sustainable transition. Financial incentives for fossil-based feedstocks—such as subsidies, tax advantages, and exemptions—should be gradually phased out to level the playing field. In parallel, the developed standards, certification schemes, and labeling systems for renewable carbon products must be unified and accepted to ensure transparency, build consumer trust, and support informed decision-making. Finally, the creation of renewable carbon-preferred product databases can further accelerate market uptake by guiding procurement and investment decisions toward sustainable alternatives [5].

Global plastic consumption is projected to continue rising in the coming decades, reaching over 1.2 billion tonnes by 2060 [3]. Based on the current commercially available plastics mix, packaging alone is expected to account for approximately 381 million metric tonnes by 2050–2060, representing more than 30% of total plastic production (Figure 1). Within this segment, polypropylene (PP) is projected to make up around 24% of the polymers used, while polyethylene terephthalate (PET) will contribute about 16% [6,7]. Packaging remains the single largest application of plastics, underscoring the importance of evaluating the role of biobased plastics within a circular economy. Transitioning this high-volume sector toward renewable and circular solutions is essential for reducing the environmental impact of plastic use [8,9,10,11,12].

## 2. Current Range of Major Plastic Packaging Solutions

A wide variety of plastic packaging solutions are currently in use, ranging from flexible to rigid formats and from mono-material to complex multilayer structures. These solutions are tailored to meet diverse functional requirements such as barrier properties, strength, flexibility, durability, and recyclability. In Figure 2, a broad selection of anonymized packaging examples illustrates this diversity, showcasing the spectrum of designs and material combinations used in the market today. Table 1 provides an overview of the most commonly used fossil-based plastic packaging materials, offering a qualitative comparison of their key characteristics—such as mechanical strength, barrier performance, recyclability, and cost. This comparison helps highlight the trade-offs involved in material selection and the challenges in transitioning to more sustainable alternatives.

**Figure 2 polymers-17-02217-f002:**
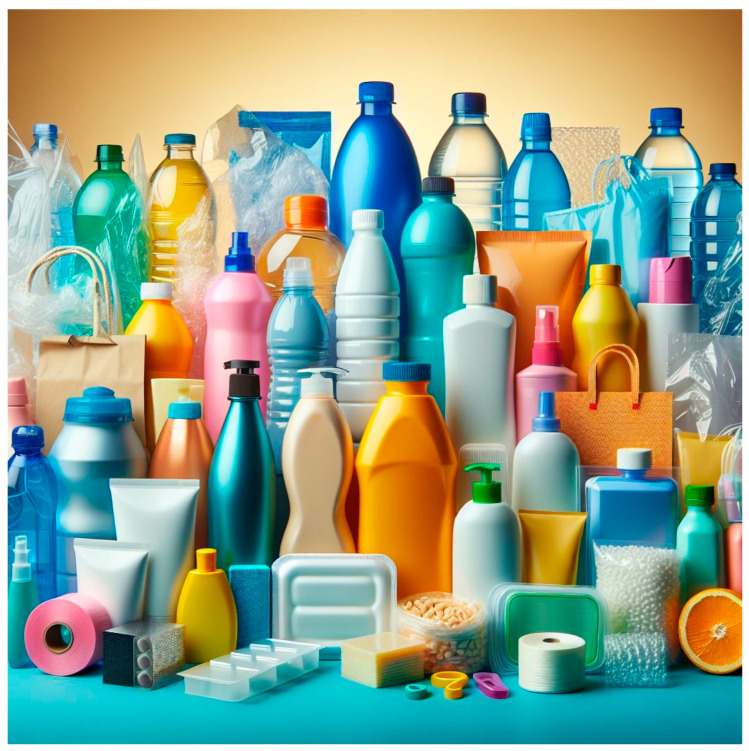
An AI-generated image [13] of the main packaging applications (water bottles, CSD bottles, shampoo and detergent bottles, food and cosmetic containers, thermoforms, bags, films, liners, caps, and closures).

Polyolefins such as low-density polyethylene (LDPE), high-density polyethylene (HDPE), and isotactic (crystalline) polypropylene (iPP) are versatile materials used in various packaging applications due to their cost-effectiveness, flexibility, moisture barrier properties, chemical and impact resistance, strength, and durability [14]. Polyethylene terephthalate (PET) is widely used in packaging for its mechanical strength, excellent O_2_ and CO_2_ barrier, heat resistance, clarity, chemical inertness, and recyclability [15]. Several other types of plastics are commonly used for packaging solutions besides polyolefins and polyesters. Polyvinyl chloride (PVC) is often used for packaging food, pharmaceuticals, and household products due to its flexibility, durability, clarity, barrier properties, and ability to be sterilized. Polystyrene (PS) is commonly used for food containers and medical packaging because of its clarity, lightweight when foamed, insulating properties, and ease of sterilization. Polycarbonate (PC) is used for packaging that requires high impact resistance and transparency, such as water bottles and food storage containers. Polyamides (PA), also known as nylon, are used for vacuum packaging and other applications requiring strong barrier properties against oxygen and CO_2_. PA also offers high chemical resistance, durability, and strength. Acrylonitrile butadiene styrene (ABS) is a versatile thermoplastic widely used in packaging applications due to its durability, impact resistance, ease of sterilization, surface properties, and simplicity of processing [16,17,18].

**Table 1 polymers-17-02217-t001:** A comparison chart for major fossil-based non-biodegradable plastic packaging solutions [16,17,18,19,20,21,22,23,24].

Property	LDPE	HDPE	iPP ^Ø^	PET	PVC ^#^	PS	PC	PA-6	ABS
Clarity	Tr *	Opaq *	Opaq	Clear	Clear	Clear	Clear	Opaq	Opaq
Tensile Strength (MPa)	4–16	21–38	25–40 (130–300 biax)	80 (190–260 biax)	25–70	30–100	55–75	78	41–45
Tensile Modulus (MPa)	100–300	400–1200	900–1500 (2200–4200 biax)	2000–4000	2500–4000	2300–4100	2300–2400	2600–3000	2100–2400
Notched Izod Impact Strength (J/m)	no break	27–1000	20–100	13–35	20–100	19–24	600–850	30–250	200–400
Chemical Resistance	High	Exc	Exc	Good	Exc	Mod	Exc	Exc	Good
Moisture Barrier (g/m^2^·day@38 °C, 90% RH 25 μm)	16–23	5–8	9–11	16–20 (biax)		10–15	Good	Exc	Mod
O_2_ Barrier (cc/m^2^·day@23 °C, 0% RH 25 μm)	7000–8500	2300–3100	2300–3100	31–93	100–300	4350–6200	100–300	20–40	500–1000 (est.)
Heat Resistance	Low	Mod	Mod	High	Mod	Low	High	High	Mod
Recyclability	Good	Good	Good	Exc	Mod	Low	Mod	Mod	Mod
Biodegradability	Low/Poor	Poor	Poor	Low	Poor	Poor	Poor	Low	Poor
Cost (€/kg)	1.00–1.50	1.00–1.50	1.00–1.50	1.00–1.50	1.00–1.50	1.00–1.50	2.00–3.00	2.00–3.00	1.50–2.00

* Tr = transparent; Mod = moderate; Exc = excellent; Opaq = opaque, biax = biaxial stretched. ^Ø^ In packaging applications, isotactic PP is most commonly used, and therefore, this column is filled for iPP. ^#^ In packaging applications, plasticized PVC is most commonly used, and therefore, this column is filled for plasticized PVC.

## 3. What Biobased Polymers Do We Want? Drop-In Versus New Functionality Polymers and Polyolefins Versus Polyesters

To make the transition from fossil-based to biobased plastics, two approaches can be followed: the production of drop-in polymers (e.g., bio-PE (100% biobased) and bio-PET (23% or 100% biobased)) [25] versus polymers with new functionality (e.g., PLA (polylactic acid) [26], PEF (polyethylene furanoate) [27,28], PHA (polyhydroxyalkanoate) [26], and PLGA (polylactic-co-glycolic acid)) [29,30]. Both approaches have their clear challenges as well as opportunities. Drop-in plastics have a relatively short time to market because they can be produced using existing infrastructure, technologies, and value chains. Bio-PET and Bio-PE, for example, are already used in packaging. In contrast, novel functionality plastics like PEF and PLA have a longer time to market due to the need for new (knowledge on) production processes, infrastructure, and the creation of their own (circular) value chains.

Drop-in materials have properties similar to their fossil-based counterparts, making them suitable for existing applications without significant modifications. Novel polymers, on the other hand, have the potential to offer unique properties. PEF, for instance, provides superior barrier properties compared to PET, with up to 10 times better oxygen barrier and around 15 times better carbon dioxide barrier [27]. PLGA is biodegradable and compostable, making it ideal for applications that have poor collection rates or are difficult to sort [29].

Cost remains a critical factor in determining the competitiveness of biobased polymers. For drop-in polymers—such as bio-ethylene, bio-terephthalic acid, and bio-MEG—achieving price parity with fossil-based counterparts is essential, as these materials do not offer inherent performance advantages. Reaching cost competitiveness requires large-scale production from the outset, which in turn demands substantial capital expenditure (CAPEX) to utilize existing industrial infrastructure and mature processing technologies [30].

In contrast, novel biobased polymers can differentiate themselves through superior or unique performance characteristics. Consequently, they can—and often must, given the absence of an established market—enter at smaller production scales. This allows for lower initial CAPEX and a more incremental market introduction. However, the cost gap between First-of-a-Kind (FOAK) and Nth-of-a-Kind (NOAK) plants is substantial, often amounting to tens of millions of euros per kilotonne of product. This disparity is driven by technical and regulatory uncertainties, lack of economies of scale, immature supply chains, and the steep learning curves associated with early-stage deployment. As technologies mature and scale increases, these costs typically decline, but the initial investment barrier remains a significant challenge for emerging biobased solutions [30].

Drop-in polymers can directly replace fossil-based PE and PET in existing packaging markets, while novel polymers need to carve out their own market spaces by offering unique properties and fitting the application requirements, such as biodegradability and superior barrier performance. Bio-PE and bio-PET face direct competition from fossil-based PE and PET, which are well-established, produced at millions of tonnes per annum per facility, and are currently cheaper than their biobased counterparts [30]. Another disadvantage of some drop-in biobased polymers, like bio-PET and PTT, is that they are only partially biobased. In contrast, novel polymers compete by offering functionalities and environmental benefits that fossil-based plastics cannot match.

Drop-ins generally comply more easily with existing regulations, as they are chemically similar to their fossil-based counterparts, while novel polymers may face more stringent regulatory scrutiny due to their novel composition, properties, and production processes. Legislation—or in some cases, the lack of specific legislation, such as the Packaging and Packaging Waste Regulation (PPWR) proposed by the European Commission [31]—can either support or hinder the development of both drop-in and novel polymers, depending on how well the regulatory framework accommodates innovation and sustainability goals. Drop-ins integrate easily into existing value chains, making adoption straightforward, while novel polymers require new value chains—challenging but offering innovation opportunities. Regarding circularity, drop-ins are fully recyclable (if recycling streams exist for their fossil equivalents) and fit into current systems. Novel polymers, due to their limited scale, need further adaptation before full integration into recycling streams [32,33]. Table 2 outlines the strengths and challenges of both drop-in and novel functionality plastics in the biobased industry. 

Polyolefin drop-ins such as bio-PE and bio-PP are hydrocarbon polymers. Producing them from oxygen-rich feedstocks like glucose (which contains 53 wt% oxygen) requires high conversion factors (CFs)—the amount of glucose needed to produce 1 ton of monomer at 100% yield. As a result, at least 3.2 tonnes of glucose are needed to produce 1 ton of monomers like ethylene, propylene, styrene, or p-xylene (the precursor for terephthalic acid). In contrast, polyester monomers derived from glucose offer significantly better yields and cost-efficiency—for example, lactic acid (CF 1.0), FDCA (2,5-furandicarboxylic acid, CF 1.1), isosorbide (CF 1.2), succinic acid (CF 1.5), and ethylene glycol (via ethanol/ethylene) (CF 1.5). From a theoretical perspective, 1 ton of glucose can generate 800 kg of PLA and 758 kg of PEF, but only 470 kg of bio-PE (all at 100% yield). When using the bioethanol route, the yield for bio-PE further drops to 312 kg per kg of glucose. These numbers clearly show that, from a yield perspective, polyesters are the preferred lowest-cost polymers over polyolefins. For that reason, it is expected—already evident from Figure 3 and Section 4—that the transition from fossil to renewable (biomass-based) feedstocks will also involve a shift from predominantly polyolefins today to primarily polyesters in the future.

In terms of properties (see Table 1 and Table 3), polyesters are generally preferable over polyolefins. From a recycling standpoint, there are clear advantages of polyesters compared to polyolefins (see also Section 7). Polyesters, such as PET and PEF, have ester linkages in their backbone and reactive end-groups and can therefore build up the chain again during recycling. This also offers advantages for chemical recycling because the ester linkages can be broken down (more) easily. This allows for efficient depolymerization and repolymerization.

Polyesters can be efficiently depolymerized into their monomers or oligomers with high purity and yield, enabling their reuse in the production of new polyester materials [6]. This depolymerization process is generally more straightforward than that of polyolefins. Mechanically recycled polyesters maintain their mechanical and thermal properties, making them suitable for various applications, including packaging, textiles, and more. Polyolefins often face challenges in maintaining their properties after mechanical recycling due to the irreversible breakages of the polymer chain. Overall, the inherent chemical properties and efficient recycling processes make polyesters a more sustainable choice compared to polyolefins [6]. Polyolefins, such as PE, are generally more persistent in nature compared to polyesters like polyethylene terephthalate (PET). Polyolefins are highly resistant to biodegradation. They primarily degrade through physical processes like photodegradation, which is slow and inefficient. This resistance makes them persist longer in the environment. Polyolefins also tend to fragment into microplastics and nanoplastics, which can accumulate and persist in ecosystems for extended periods [34].

## 4. Biobased Plastics, State of the Art

Each year, European Bioplastics and the nova Institute publish an update on global bioplastic production (Figure 4 and Figure 5). In this publication, European Bioplastics uses the term “bioplastic” to refer to biobased and/or biodegradable plastics, even if the biodegradable plastics are fossil-based. Many scientists define bioplastics (and associated biopolymers) more narrowly as polymers produced by organisms in nature. Therefore, from this perspective, we will use the terms “biobased polymer” and “biobased plastic”, as shown in Figure 3 (adapted from Gruter [2]). Figure 3 also helps clarify the distinction between biobased and biodegradable polymers, which are often mistakenly conflated [35].

Almost half of the production shown in Figure 4 consists of biobased, non-biodegradable plastics. For some biobased plastics, not all building blocks are currently biobased (e.g., PTT, bio-PET, PBAT). Alongside fully biobased plastics, bio-attributed variants of conventional polymers—developed through the mass balance approach—are gaining traction in the market. These include bio-attributed polyethylene (bio-PE), polypropylene (bio-PP), and polyvinyl chloride (bio-PVC), among others. In this process, renewable feedstocks are co-processed with fossil-based raw materials, and the resulting biobased content is assigned to specific product batches using a certified accounting system.

However, the majority of biobased plastics production currently takes place outside Europe (Figure 5). The “EU policy framework on bioplastics” [37] aims to shift this balance by fostering local production capacities.

## 5. Biobased Packaging Solutions

A wide range of biobased polymers holds potential for use in packaging applications. However, their levels of technological maturity vary significantly. While some have already reached commercial-scale production, others remain at early stages of development with low Technology Readiness Levels (TRLs).

In Section 5 and Section 6, we focus on the most advanced solutions—those with a TRL of 5 or higher. Section 5 explores biobased plastics, while Section 6 addresses biodegradable plastics. To highlight the European dimension of this emerging field, companies based in Europe that are active in these areas are indicated in **bold**.

a.PEF

Polyethylene furanoate (PEF) is a biobased polymer derived from renewable plant sugars, specifically 2,5-furandicarboxylic acid (FDCA) and ethylene glycol [27,38]. It is considered a promising alternative to conventional plastics like PET due to its superior properties and environmental benefits. PEF has excellent barrier properties against gases such as oxygen, carbon dioxide, and water vapor [39,40,41]. It has up to ten times lower oxygen permeability and around 15 times lower carbon dioxide permeability compared to PET, making it highly effective for food and beverage packaging [27]. PEF exhibits a melting temperature (Tm) of approximately 210–230 °C and a glass transition temperature (Tg) of around 88 °C. It also has good thermal stability, with decomposition temperatures around 350 °C. PEF offers high mechanical strength and rigidity, which makes it suitable for various packaging applications. Most of its mechanical properties are even better than those of PET. PEF is resistant to many chemicals, which enhances its durability and suitability for packaging applications. Being derived from renewable resources, PEF is a more sustainable option compared to petroleum-based plastics. It is also recyclable back into recycled PEF using existing mechanical recycling assets designed for PET recycling, contributing to a circular economy [27]. PEF is used for bottles and containers for water, juices, carbonated drinks, and alcoholic beverages. Its high barrier properties help retain carbonation and extend the shelf life of these beverages. PEF is suitable for packaging both fresh and dry foods. Its barrier properties against oxygen and moisture help maintain freshness and extend the shelf life of food products. PEF is also used in flexible (multilayer) films that protect processed meat, snacks, cheese, and other food items from oxygen and moisture. In addition, it is used in cosmetic packaging due to its transparency and aesthetic appeal, providing a clear and attractive look for beauty products. These properties make PEF an attractive material for applications ranging from food and beverage packaging to textiles and other consumer goods. Major players are **Avantium**, Sugar Energy, and GS Biomats.

b.Bio-PE

Biobased polyethylene (bio-PE) is chemically identical to fossil-based PE and predominantly produced from sugarcane with ethanol as the intermediate [25]. Bio-PE is primarily used in packaging applications. This includes both flexible and rigid packaging for various industries, such as Tetrapack Rex^®^ plant-based cartons. A major player is Braskem under the tradename “I’m green^TM^”.

c.Bio-PP

Biobased polypropylene (bio-PP) is chemically identical to fossil-based PP. Often, the mass balance approach is used to track and allocate the amount of biobased content. Several routes can be used to produce bio-PP; one example is using bioethanol derived from renewable resources like sugarcane or via waste and residual oils [25]. Bio-PP is primarily used in food packaging applications such as containers, bottles, and films. Major players are Braskem, **Neste/LyondellBasell,** and **Neste/Borealis**.

d.Bio-PA

Biobased polyamides are predominantly used in outlets outside the packaging area, such as the automotive industry, textiles, electronics, sporting goods, and consumer goods. Biobased polyamides are increasingly used in various packaging applications due to their sustainability and excellent performance characteristics [42]. Biobased polyamides offer excellent barrier properties against gases and moisture, making them ideal for food packaging. They help extend the shelf life of food products while being environmentally friendly. Due to their biocompatibility, biobased polyamides are suitable for packaging medical devices and pharmaceuticals. They ensure the safety and integrity of medical products. Polyamides are also used in flexible and cosmetic packaging. Major producers include **Arkema**, **Evonik**, **BASF**, **Envalior**, Celanese Corp., and Mitsubishi.

e.Bio-PET

Biobased polyethylene terephthalate (bio-PET) is chemically identical to fossil-based PET. Currently, only the monoethyleneglycol (MEG) part is biobased, resulting in a bio-PET with ~23% biobased content. MEG is currently predominantly produced from sugarcane with ethanol and ethylene intermediates. In the near future, MEG produced in Europe from lignocellulosic biomass will also enter the market [25]. Many players are working on bio-PTA, but so far, it is inefficient to produce bio-PTA from renewable resources, and therefore, the economics remain challenging. An additional challenge is that isophthalic acid (IPA) is commonly used in bottle-grade PET as a co-monomer to modify the crystallization rate. Bio-IPA is not available or expected to become available in the near future. Major players for biobased MEG are Indian Glycols and **UPM**.

f.Bio-PTT

Biobased polytrimethylene terephthalate (bio-PTT) is prepared by the esterification of 1,3-propanediol (1,3-PDO) with terephthalic acid. 1,3-PDO is derived from plant-based resources, such as corn glucose. Bio-PTT is known for its excellent durability, high melting point, and good chemical resistance, making it suitable for a variety of applications, including textiles, automotive parts, and packaging. In packaging applications, bio-PTT is not widely used but can offer several advantages, including good barrier properties, durability, thermal stability, and chemical resistance [43]. Production costs of bio-PTT are currently higher than bio-PET. Some of the major producers of bio-PTT include Covation, Zhangjiagang Glory Biomaterial, and Shenghong Group.

## 6. Biodegradable Packaging Solutions

Biodegradability in packaging materials can currently be certified under six distinct environmental conditions: marine environments, freshwater, soil, home composting, anaerobic digestion, and industrial composting. Some polymers demonstrate biodegradability across all these conditions, while others are limited to specific environments—most commonly industrial composting and thermophilic anaerobic digestion [44]. It is important to note that the biodegradability of a plastic product depends not only on the polymer itself but also on the complete formulation. For a plastic to be truly biodegradable, all additives and organic fillers used in the material must also be biodegradable (Nova Institute). Table 1 and Table 3 highlight the key properties, common applications, barrier properties, and cost levels of various plastic and bioplastic types, providing insights into their strengths and weaknesses for different packaging solutions.

a.PLA

Polylactic acid (PLA) is a slow-biodegrading thermoplastic derived from renewable resources like cornstarch or sugarcane [26,44]. PLA is environmentally friendly, as its production uses less energy and generates fewer greenhouse gases compared to conventional plastics [45,46]. However, it has limitations, such as a low glass transition temperature, typically around 55–60 °C. This means it can deform or lose its shape when exposed to high temperatures, making it unsuitable for applications requiring heat resistance. PLA is generally more brittle and has lower impact strength compared to conventional plastics like PET or PP. This limits its use in applications requiring high durability and toughness. PLA can degrade when exposed to UV light and moisture over time. While this can be beneficial for biodegradability, it can also limit the lifespan of products made from PLA. PLA requires specific conditions to biodegrade, such as those found in industrial composting facilities. PLA is widely used for containers, drinking cups, sundae and salad cups, overwraps, and blister packages. It is especially suitable for short shelf-life products like fruits and vegetables. PLA is also used for bottles, jars, and tubes for beauty and personal care products. PLA is used for packaging medical devices and supplies due to its biocompatibility and biodegradability, and PLA is used for clamshells and trays, providing a sustainable alternative to traditional plastics. China’s biobased polymer industry is experiencing rapid growth despite being in its early stages, largely propelled by policy incentives. Projections suggest that the industry will expand substantially, reaching 2.53 million tonnes by 2026, up from 765,000 tonnes in 2023, representing a notable CAGR of approximately 49% [37]. Major players are NatureWorks, Anhui BBC Biochemical & **Futerro**, **TotalEnergies-Corbion**, Zhejiang Hisun Biomaterials Co., and Emirates Biotech.

b.PHA

Polyhydroxyalkanoates (PHAs) are a promising class of biobased plastics for packaging applications due to their biodegradability (nova biodegradability), compostability, and versatility [26]. PHAs are produced by microbial fermentation using renewable feedstocks like sugar and starch or waste streams containing fermentable biomass [26,47]. There are two primary forms of PHA:Amorphous PHA (aPHA): Soft and rubbery, used as a modifier to improve the properties of other polymers like polylactic acid (PLA).Semi-crystalline PHA (scPHA): Offers greater stiffness and high-heat stability, suitable for rigid packaging.

To improve the applicability of PHA, production costs need to come down, as it is generally more expensive compared to conventional plastics. The higher costs are due to the expenses associated with microbial fermentation, feedstock, and downstream processing (isolating and purifying the intracellular PHA). A major challenge is that PHA requires specific processing conditions, which can be a limitation for some applications [47]. This includes the need for precise temperature and humidity control during processing.

While PHAs have good mechanical properties, they may not always match those of conventional plastics. Blending PHAs with other biopolymers or additives can help improve their performance [26]. Another challenge is to improve the recyclability of PHA by integrating mechanical, chemical, and biological recycling pathways [48]. Addressing these challenges is key to the broader adoption of PHA in packaging applications. Major players in the PHA market include **Bio-on SpA**, Bluepha, CJ Biomaterials, Danimer Scientific (filed for bankruptcy), Kaneka Co., and **Paques Biomaterials**.

c.Thermoplastic starch (TPS)

Thermoplastic starch (TPS) is a biodegradable polymer derived from natural starch sources like corn and wheat. It is created by plasticizing starch with water and plasticizers (such as glycerol or sorbitol) under heat and shear, transforming it into a thermoplastic material. TPS is fully biodegradable, making it an environmentally friendly alternative to conventional plastics [26,44,49]. TPS exhibits good flexibility, which is beneficial for various packaging applications. TPS has inherent moisture sensitivity, which can be both an advantage and a limitation depending on the application. TPS provides moderate barrier properties against gases and moisture, suitable for short-term food packaging. TPS can be blended with other polymers to enhance its mechanical properties and processability. TPS is used for compostable food packaging, such as bags for fruits and vegetables, snack packaging, and cereal boxes. TPS is also employed in biodegradable mulch films and greenhouse films, helping retain soil moisture and suppress weeds [49]. TPS is suitable for single-use packaging applications where biodegradability is crucial. Despite its advantages, TPS requires specific processing conditions and may need to be blended with other polymers to improve its performance for certain applications. Major players are **Novamont**, **Biotec**, **Rodenburg Biopolymers**, and **Agrana.**

d.Cellulose derivatives (CDs)

Cellulose derivatives (CDs) are widely used in packaging due to their biodegradability, renewability, and excellent barrier properties [50]. Important cellulose derivatives include cellophane, which is a transparent, biodegradable film made from regenerated cellulose, carboxymethyl cellulose (CMC), and methyl/ethylcellulose (MC/EC), used for its film-forming and thickening properties. It is often incorporated into edible films and coatings for food packaging. Hydroxypropyl methylcellulose (HPMC) is known for its excellent film-forming abilities and is used in pharmaceutical and food packaging. Cellulose acetate is a derivative used for its strength and transparency. It is commonly used in packaging films and coatings. These cellulose derivatives are chosen based on their specific properties, such as barrier performance, biodegradability, and mechanical strength. In particular, in smart/intelligent packaging, cellulose derivatives can be used in packaging films that monitor the freshness and quality of food products through colorimetric indicators or sensors. In active packaging, chemically modified cellulose can be used to create packaging that actively interacts with the food to extend its shelf life by controlling moisture, oxygen, and other factors. These cellulose derivatives bring unique properties [50,51]. Major players are Dow, Ashland, **Borregaard**, **Wacker Chemie**, CP Kelco, and Shin-Etsu.

e.PBAT and PBxF

Polybutylene adipate terephthalate (PBAT) is a biodegradable and compostable polymer widely used in packaging applications due to its flexibility, moisture resistance, and eco-friendly properties [26,44,52]. PBAT is ideal for compostable food packaging, such as fruit and vegetable bags, snack packaging, and cereal boxes. Its ability to provide a good barrier against fats and liquids helps extend the shelf life of food products. Additionally, PBAT is used for biodegradable mulch films, greenhouse films, and silage covers, helping retain soil moisture and suppress weeds. PBAT can be blended with more rigid polyesters to improve its mechanical properties. The current biobased share is about 34% and based on biobased 1,4-butanediol. **BASF** has commercialized ecovio, a PBAT/PLA blend [53].

It has been reported that polybutylene adipate furanoate (PBAF) has even superior properties compared to PBAT [54]. Fully biobased equivalents, such as PBxF (where x represents a biobased diacid like succinic, azelaic, or sebacic acid), are expected to be prepared with similar properties to PBAT or PBAF [54]. The BASF-led Horizon Europe project ReBiolution is investigating the optimal PBxF formulation and its blends for food packaging and mulch films. Currently, there are no major players producing PBxF.

f.PBS

Polybutylene succinate (PBS) is a slowly biodegradable aliphatic polyester with properties that are comparable to polypropylene [44,55]. The scope of PBS application fields is still growing. In the packaging field, PBS can be processed into films, bags, or boxes for both food and cosmetic packaging. Other applications of PBS are disposable products such as tableware or medical articles. In agriculture, PBS finds interest in the fabrication of mulching films or delayed-release materials for pesticides and fertilizers. PBS is also applied in fishery (for fishing nets), forestry, civil engineering, or other fields in which the recovery and recycling of materials after use is problematic. In the medical field, PBS is used as a biodegradable drug encapsulation system. Biobased PBS currently has a 57% biobased content due to the use of biobased succinic acid. Mitsubishi Chemical is the main producer of biobased PBS. In addition, there are several other Asian producers of fossil-based PBS.

**Table 3 polymers-17-02217-t003:** Commercial biobased plastic packaging solutions. Properties derived from the following references: [22,26,27,30,54,55,56,57,58,59,60,61,62].

Property	Bio-PE	Bio-PET	PLA	PLGA 9/91	PBS	TPS	PHA	CD	PEF	PBAT ^#^
Clarity	Tr *	Clear	Clear	Clear	Tr	Tr	Tr	Tr	Clear	Tr
Tensile Strength (MPa)	4–28	80 (190–260 biax)	60–70	15–18	30–50	5–20	20–40	Mod	50–70	10–35
Tensile Modulus (MPa)	410–1400	2000–4000	2700–3500	25–40	300–500	50–200	1000–3500	Mod	3000	100–200
Notched Izod Impact Strength (J/m)	27–>1000	13–35	2000–4000	low	4000–50,000	Very Low	2000–10,000	Mod	Mod.	20,000–50,000
Chemical Resistance	Exc *	Good	Mod	Mod	Mod	Low	Mod	Mod	Exc	Mod
Moisture Barrier (g/m^2^·day@38 °C, 90% RH) 25 μm)	5–23	16–20	40–60	3	20–30	50–80	10–20	Mod	14	40–60
Heat Resistance	Mod	High	Mod	Mod	Mod	Low	Low	Mod	High	Mod
Recyclability	Mod	High	Mod	Mod	Mod	Low	Low/Mod	High	High	High
O_2_ Barrier (cc/m^2^·day@23 °C, 0% RH 25 μm)	2300–8500	31–93	1500–3000	5	1000–2000	3000–5000	100–500	Low	1–10	2000–4000
Biodegradability	Poor	Poor	Mod	High	High	High	Good/High	Low	Low	Mod/Good
Cost (€/kg)	2.00–3.00	1.50–2.00 (20% bio)	>4.00	5.00–10.00	>4.00	Mod	>4.00	High	10.00 ^ø^	3.00–4.00

* Tr = translucent; Mod = moderate; Exc = excellent; biax = biaxial stretched. ^#^ Ecovio (BASF) is a PBAT/PLA blend that integrates the good properties of both materials. ^ø^ Cost from the flagship plant, future scale-up will reduce price to 4–5 €/kg.

g.Aliphatic polycarbonates (APCs)

Aliphatic polycarbonates (APCs) are a big family of biodegradable polymers known for their biocompatibility and controlled biodegradability. They are synthesized from aliphatic diols and carbonates, such as dimethyl carbonate (DMC), through processes like polycondensation or ring-opening polymerization [63]. APCs are used in several packaging applications due to their biocompatibility, especially for food, cosmetic, and pharmaceutical packaging. Major players are Asahi Kasei, **Teysha Technologies**, Mitsubishi, and **Covestro AG.**

h.Novel biobased polymers

Poly(lactic-co-glycolic acid) (PLGA) is a biodegradable copolymer that is currently predominantly used for medical applications [64] but has also garnered significant R&D attention for packaging applications due to its biodegradability, biocompatibility, sustainability, and excellent mechanical and barrier properties [30,65]. The mechanical and physicochemical properties of PLGA are strongly determined by the ratio of the monomers. Most commercial PLGA grades currently available contain mainly lactide (between 50 and 95 mol%), with the exception of the absorbable suture material brand Vicryl (90 mol% glycolide). Copolymers with between 50 and 85 mol% lactide exhibit a tensile strength between 41 and 55 MPa and a tensile modulus between 1 and 4.3 GPa. These values tend to be close to those reported for PLA and considerably lower than those for PGA. Important properties (e.g., barrier properties, thermal stability, and crystallization behavior) useful for determining their potential for a wider range of applications were recently reported [30]. Major players in the PLGA market include **Evonik**, **Corbion NV**, Ashland, and Mitsui Chemicals. Avantium is developing a process to produce glycolic acid from CO_2_ [66] and PLGA via direct polycondensation of glycolic and lactic acid [67].

Renewable polyesters with a good balance between impact strength and elastic modulus are rare, especially when combined with a high glass transition temperature (Tg). Meeting such performance would enable the substitution of polymers like acrylonitrile butadiene styrene (ABS) and polycarbonate with chemically recyclable polyesters from biobased, CO_2_-based, or recycled sources for rigid packaging applications [68]. High isosorbide content polyesters with high molecular weights that display good performance for the three targeted parameters include polyisosorbide-co-butanediol terephthalate (PIBT) with 1,4-butanediol, poly-isosorbide-co-cyclohexane di-methanol terephthalate (PICT) with 1,4-cyclohexanedimethanol (CHDM), and poly-isosorbide-co-propanediol terephthalate (PIPT) with 1,3-propanediol. For replacing ABS, PICT and PIBT seem to be excellent candidates for high-performance applications while maintaining high thermal stability. PICT also outperforms ABS significantly in terms of impact strength and Tg while maintaining a relatively high elastic modulus at 2 GPa [69].

Polyesters composed of certain combinations of bulky monomers like isosorbide, CHDM, and Tritan’s monomer TMCBD (2,2,4,4-Tetramethyl-1,3-cyclobutanediol) tend to display a relatively high Tg, high impact strength, and relatively high stiffness. Combined with an efficient synthesis route like the reactive solvent method, these monomers are valuable for compositions of tough, rigid, high-Tg polyesters [70]. Producing high-molecular-weight isosorbide-based (co)-polyoxalates (PISOXs) using diguaiacyl oxalate facilitates the generation of high molecular weights in short reaction periods, even in the absence of a catalyst. Compared to currently available biobased plastics, PISOX displays superior mechanical, thermal, and barrier properties. PISOX stands out for its rapid biodegradability in both soil and seawater. It is remarkable that such a thermally and mechanically strong material degrades in a matter of months, which holds potential for specific applications [71].

Table 4 provides a comprehensive overview of the most widely used fossil-based plastics across various packaging applications, ranging from food containers and beverage bottles to flexible films and industrial wraps. These conventional materials have long dominated the packaging sector due to their versatility, durability, and cost-effectiveness. However, growing environmental concerns and regulatory pressures have spurred interest in identifying sustainable alternatives. The table also highlights promising biobased counterparts—such as PLA, PEF, and PHA—that offer comparable functionality while reducing reliance on fossil resources. By juxtaposing traditional plastics with their renewable substitutes, Table 4 serves as a valuable reference for stakeholders aiming to transition toward more circular and eco-friendly packaging solutions.

## 7. Recyclability of Biobased Packaging Solutions

Both mechanical and chemical recycling of drop-in plastics (e.g., bio-PE, bio-PP, bio-PET) follow the same processes as their fossil-based counterparts. In this section, we will first discuss conventional mechanical and chemical recycling and then address the opportunities and challenges of recycling biobased plastics [78]. In Figure 6, the global polymer flows are given. It is clear that recycling is still a minor component in the end-of-life options [79].

### 7.1. Mechanical Recycling

Mechanical recycling of polyesters and polyolefins shares some similarities but also has distinct differences due to their chemical structures and properties. Both involve similar basic steps: collection, sorting, cleaning, shredding, melting, and reprocessing into new products. Every melt processing step breaks down the polymer chain. For polyesters, this process is reversible, while for polyolefins, this is irreversible [80].

Polyesters are more easily decontaminated than polyolefins, allowing for higher-quality recycled products even from less clean input materials. Recycled polyester (rPET) is commonly used in textiles, packaging, and high-demand technical applications, while recycled polyolefins are often used in lower-value applications like construction materials, automotive parts, and non-food packaging [81].

Recycling processes in general, and mechanical recycling processes in particular, are highly dependent on the quality of the input feedstock. Rigid plastic packaging typically yields higher-quality recyclate than flexible packaging due to its lower proportion of multilayer structures and higher mass-to-volume ratio, which facilitates more efficient sorting. While several technologies are under development to enable the recycling of complex multilayer materials [82], their economic feasibility remains a significant challenge. Advancing these technologies and improving feedstock quality are both critical to scaling effective recycling systems.

**Figure 6 polymers-17-02217-f006:**
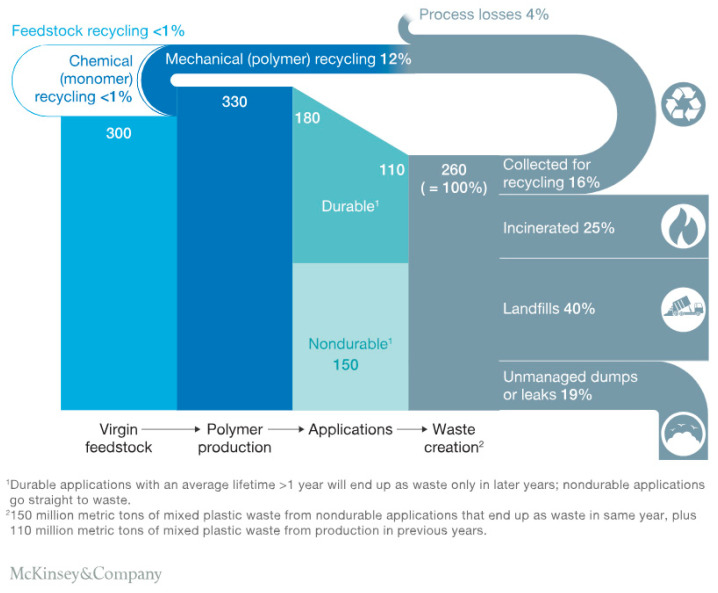
Global polymer flows in millions of metric tonnes per annum, adapted from Hundertmark et al. [79].

#### Mechanical Recycling of Biobased Plastics

Mechanical recycling of biobased plastics offers its own opportunities and challenges [78]. It supports the circular economy by keeping sustainable carbon materials in the loop as long as possible, thereby reducing the need for virgin materials, further lowering the carbon footprint, and contributing to sustainability goals [83]. Eventually, when these materials reach their end-of-life—such as through incineration—the CO_2_ released is largely offset by the carbon previously captured from the atmosphere by the biomass used to produce them. This carbon-neutral cycle reinforces the environmental benefits of biobased plastics when paired with effective recycling strategies.

Biobased plastics come in various types, such as PEF, PLA, PHA, TPS, and CD, each with distinct properties and recycling requirements. This diversity complicates the sorting and recycling process. Some biobased plastics degrade more quickly than conventional plastics, leading to a loss in mechanical properties like tensile strength after each recycling cycle. The existing recycling infrastructure is primarily designed for high-volume fossil-based plastics, and adapting it to handle small volumes of innovative biobased materials efficiently requires investment and adjustments in the recycling infrastructure. Sorting one-to-many also significantly slows down sorting versus positive (or negative) sorting (one-to-two: keep and discard). However, innovations in sorting and recycling technologies, such as near-infrared (NIR) sorting and HolyGrail 2.0 watermarking [84], can improve the efficiency and effectiveness of (biobased) plastic recycling.

However, there is a clear chicken-and-egg issue when considering the recycling of innovative biobased plastics. While the volumes of biobased plastics are low, there is no strong incentive for recyclers to adapt their infrastructure and sort out the biobased plastics. The lack of recycling infrastructure hampers the introduction of biobased plastics. An interesting case is presented with the biobased plastic PEF, which has been awarded an interim recycling endorsement from EPBP (European PET Bottle Platform) up to a maximum market penetration of 2% [85]. The interim endorsement is based on a thorough assessment that shows that, for up to 2% of PEF in the rPET stream, there is no negative impact on haze, color, and quality. As a result, sorting errors involving PEF do not compromise rPET quality, enabling the gradual development of a dedicated PEF recycling stream even at low volumes. Similar to how recyclers manage colored PET campaigns, they could run occasional PEF campaigns as volumes increase. Moreover, a recent study by a leading recycler and preform producer suggests that small amounts of PEF in rPET may actually enhance its quality by slowing crystallization, which can improve processing behavior [unpublished results]. In this context, PEF becomes an asset rather than a contaminant—a rare case where a bioplastic can positively contribute to existing recycling streams.

To ensure the circular introduction of innovative biobased plastics and avoid the loss of valuable material, a harmonized, EU-regulated approach for the circular introduction of innovative materials is recommended. Facilitating the integration of innovative biobased plastic into existing recycling and waste management systems could be facilitated via two main routes.

Innovative biobased plastics that are (limited) compatible with current recycling streams: Based on a thorough assessment, x percentage of the innovative material can be allowed, sorted, and recycled, together with the incumbent material, without harming the quality of the recycling stream. A dedicated recycling stream can be created via, for example, NIR (near infrared) sorting once the critical mass is reached.Innovative biobased plastics that are not compatible with current recycling streams: A bulk stream can be created that sorts all non-compatible biobased materials together. This dedicated mix of biobased materials goes to a specialized, biobased recycler that further sorts and recycles the biobased materials.

Governments and regulatory bodies are increasingly supporting recycling initiatives, which can be an opportunity to provide incentives and funding for biobased plastic recycling projects. Mechanical recycling of biobased plastics presents both challenges and opportunities, but with continued innovation and support, it can become a more viable and sustainable option.

### 7.2. Chemical Recycling

As of 2025, the commercialization of chemical (advanced) recycling of plastics is progressing, but the landscape remains mixed with both notable advancements and significant challenges. Installed global capacity for chemical recycling reached nearly 1 million tonnes per year by the end of 2024, with projections to exceed 3 million tonnes/year based on announced projects. Technologies in use include pyrolysis, solvolysis, gasification, and depolymerization, with pyrolysis accounting for a significant share of current capacity [86]. However, economic viability remains a major hurdle. Many projects have faced delays or shutdowns due to high costs, technical complexity, and fierce competition with low-cost virgin plastics [86]. Chemical recycling of polyolefins, such as PE and PP, typically involves processes like pyrolysis (followed by hydrotreating to increase ethylene and propylene yields) and gasification [87,88]. The yields from these processes can vary based on several factors, including the type of polyolefin, the specific recycling method, and the reaction conditions. Pyrolysis is a common method for chemically recycling polyolefins, involving heating the plastic waste in the absence of oxygen to break it down into smaller molecules. The resulting pyrolysis oil can be added to the cracker to produce the building blocks for new polyolefins. The carbon efficiency of these processes is not expected to be higher than 50%. Gasification converts polyolefin waste into syngas (a mixture of hydrogen and carbon monoxide) at high temperatures [87,88].

Chemical recycling of polyesters, particularly polyethylene terephthalate (PET), is an advanced method that breaks down the polymer selectively into its monomers or oligomers, which can then be re-polymerized into new virgin-quality polyester products. Several processes are available, including hydrolysis, enzymatic hydrolysis, glycolysis, and methanolysis. Currently, it is not yet clear which process is the most cost-effective, as this depends on various factors, including energy consumption, catalyst costs, feedstock specifications (impurity tolerance), and operational efficiency. The major advantage of chemical recycling over mechanical recycling is that chemical recycling can produce high-purity monomers without contaminants, which can be used to create high-quality new polyester products [88,89]. Biobased polyesters such as PEF, PLA, and PBS can be chemically recycled via hydrolysis or alcoholysis into monomers that can be repolymerized [88,90,91].

### 7.3. Advantages of Closed-Loop Recycling

Closed-loop recycling is a process in which a product or material is recycled into a new product without losing its original properties. In practice, however, there are relatively few examples of closed-loop recycling, primarily due to challenges in collection and sorting [92,93,94]. For food contact articles, specific hurdles include the need to remove food residues from the plastic waste stream and the requirement for a minimum proportion of food-grade materials in the recyclers’ feedstock. The most prominent example of large-scale closed-loop recycling is found in PET bottle recycling [92]. In countries with deposit return systems, the quality of the collected bales can be effectively controlled, making closed-loop recycling feasible at recycling rates well above 50%. These deposit systems are designed to keep materials in circulation for as long as possible by minimizing contamination and preventing the mixing of food-grade plastics with other waste streams. This results in a high-quality feedstock for recyclers and reduces the demand for virgin raw materials. Additionally, closed-loop recycled materials often require less energy and labor to process than new materials, potentially leading to significant cost savings and improved economic efficiency, although overall cost efficiency compared to virgin materials remains a challenge [93,94]. However, as closed-loop recycling schemes become more successful and recycling rates increase, the limitations of the recycled resin become more apparent. With each recycling cycle, the average number of loops the resin has undergone increases [95], leading to a higher concentration of non-intentionally added substances (NIAS), contaminants, and discoloration of the resin. In this context, depolymerization techniques show promise as a complementary approach, offering the potential to restore recycled resins to virgin-quality standards.

## 8. Sustainability of Biobased Plastics

A central question in evaluating the sustainability of biobased plastics is whether sufficient biomass will be available to replace fossil carbon in the chemicals and derived materials sector. Specifically, can 20% of the sector’s global carbon demand be met by biomass by 2050? This 20% target has emerged as a benchmark for what is considered both ambitious and realistically achievable within environmental and economic constraints. A study commissioned by RCI and the Biobased Industries Consortium (BIC) concluded that this target is indeed attainable and sustainable [6]. The analysis found that agricultural residues and woody biomass can be mobilized at scale without jeopardizing food and feed security or competing with biofuel production [6,96]. Under moderately advanced technological development scenarios—deemed the most likely pathways—the 20% substitution level can be reached effectively. However, the study also cautions that aiming significantly beyond this threshold would be impractical under current agricultural practices and biofuel policy frameworks, as it could lead to unsustainable land use pressures and resource competition [6].

Another important approach to evaluating the sustainability of (biobased) plastics is the Life Cycle Assessment (LCA), a comprehensive methodology that assesses the environmental impacts of a product across its entire life cycle—from raw material extraction through production, use, and end-of-life disposal. LCA studies consistently demonstrate that biobased plastics generally exhibit a significantly lower carbon footprint compared to their fossil-based counterparts [26,27,45,46,58,97,98,99]. A substantial portion of the climate impact of these materials is associated with the embodied carbon that is eventually released at the product’s end-of-life [99]. This reduction is primarily due to the use of renewable raw materials, which absorb atmospheric CO_2_ during their growth phase. However, a notable share of the climate change impact associated with these materials arises from the release of embodied biogenic carbon at the product’s end-of-life [99].

Most LCA standards and guidelines have an established consensus on the importance of explicitly modeling, tracking, and quantifying biogenic carbon flows using characterization factors of −1 for CO_2_ uptake and +1 for CO_2_ emissions (the so-called −1/+1 approach). In contrast, other methodologies—such as the 0/0 approach adopted in the Product Environmental Footprint (PEF) framework [99,100]—do not explicitly account for biogenic CO_2_ uptake or release, on the premise that these are part of a balanced carbon cycle, in which the CO_2_ absorbed during biomass growth is eventually re-emitted into the atmosphere. While the 0/0 approach may be suitable for modeling closed carbon cycles, the −1/+1 approach provides a more accurate representation of the environmental performance of biobased materials, particularly when comparing them to fossil-based alternatives [101].

The release of embodied carbon-associated emissions can be mitigated by increasing recycling rates, as highlighted by the papers of Stegmann [98] and Zuiderveen [99]. Moreover, it is increasingly recognized that product sustainability assessments should integrate considerations of reuse, recycling, and remanufacturing. In this context, closer alignment between environmental impact assessments and circularity metrics—such as Circularity Indicators—could enhance the robustness of sustainability evaluations [99,102].

The production of biobased plastics can be more energy-efficient than traditional plastics, although this varies depending on the type of bioplastic and the production process used. Scaling the industry is very likely to further improve energy efficiency. Many biobased plastics are designed to biodegrade under specific conditions, reducing long-term environmental pollution. However, the actual biodegradation rate depends on the environment, such as industrial composting facilities [103].

End-of-life management is a crucial aspect of bioplastic sustainability. Proper recycling and disposal infrastructure are critical for realizing the environmental benefits of biobased plastics. Without appropriate facilities, biobased plastics may not be properly recycled, contributing to pollution, and may not degrade as intended. The overall environmental impact of biobased plastics is highly context-specific. Factors such as agricultural practices, energy sources, and waste management systems play significant roles in determining their sustainability.

In summary, while biobased plastics offer several environmental advantages over conventional plastics, their sustainability is influenced by various factors throughout their life cycle. Continued advancements in production technologies, improved waste management infrastructure, and responsible agricultural practices are essential to maximizing the benefits of biobased plastics [6].

## 9. Biobased/Biodegradable Packaging Solutions Versus the Formation of Microplastics

Microplastics from plastic packaging solutions are primarily formed through two mechanisms: fragmentation and (bio-)chemical reactions [34,104]. Fragmentation occurs due to physical processes like UV radiation, mechanical wear, and abrasion, causing larger plastic items (e.g., bottles, bags) to break into smaller particles [105]. (Bio-)chemical reactions, such as oxidation (often accelerated by (UV) light) and hydrolysis (polyesters), further degrade plastics into micro-sized particles. Microplastics are pervasive in oceans, rivers, and soils, contributing to widespread environmental pollution [106]. They are ingested by marine organisms, leading to physical harm, reduced feeding, and potential death [107]. Microplastics have been found in human tissues, including the lungs and digestive system, posing potential health risks such as inflammation, oxidative stress, and endocrine disruption [34,106]. They can enter the food chain through seafood and other contaminated food sources, posing risks to human health. Microplastics are present in the air we breathe, contributing to respiratory issues and other health problems [105]. They can leach harmful chemicals, such as bisphenol A (BPA) and phthalates, which are known to have toxic effects [34,106]. Microplastics can act as carriers for pathogens and harmful microorganisms, potentially spreading diseases. Their persistence complicates waste management and recycling efforts, as they are difficult to remove from the environment. The environmental and health impacts of microplastics can lead to significant economic costs, including healthcare expenses and loss of biodiversity [107]. There is a lack of comprehensive regulations and standards to address microplastic pollution, making it challenging to mitigate their impact effectively [106].

Both polyolefins (PE, PP, PS) and polyesters (PET, PTT, PEF) can form microplastics, but the rate and extent of degradation vary based on environmental conditions and specific polymer properties [27,34]. Generally, polyolefins like PE are highly resistant to natural degradation, taking hundreds of years to break down in the environment. Polyolefins primarily degrade through physical processes such as UV radiation and mechanical wear, fragmenting into microplastics. PET also degrades slowly but can break down slightly faster than PE under certain conditions, still taking decades to centuries to fully degrade. PET degradation involves both physical and chemical processes, with microorganisms like bacteria and fungi enzymatically breaking down PET, although this process is slow and dependent on factors like surface area and crystallinity. The non-biodegradable biobased polyester PEF degrades much faster than PET [27]. Modifying fossil-based or biobased polyester by using longer diols (e.g., butanediol) and aliphatic diacids (e.g., adipic or sebacic acid) makes these polyesters biodegradable. Other polyesters, such as PLA, PLGA, and PHA, are biodegradable under industrial composting conditions (PLA) as well as home composting conditions (PLGA, PHA) [44]. Biodegradable plastics do not contribute to the formation of persistent microplastics [108].

However, it is important to recognize the inherent contradiction between rapid biodegradation and optimal plastic performance during use. Characteristics such as excellent barrier properties, moisture resistance, high strength, impact resistance, and processability typically require plastics that are not highly biodegradable, although for certain applications, highly biodegradable biobased plastics such as PLGA are very suitable. In this context, there are strong indications that biobased polyesters such as PEF, PLA, PLGA, and PHA offer the best of both worlds. They outperform their fossil-based counterparts on some important characteristics while also exhibiting significantly faster biodegradability than PET [27].

## 10. Safe and Sustainable by Design (SSbD) of Plastic Packaging Solutions

SSbD is a framework aimed at guiding the innovation process for chemicals and materials to ensure they are safe for humans and the environment while also being sustainable throughout their life cycle. In packaging applications, SSbD principles help in designing materials that minimize environmental impact during sourcing, production, use, reuse, and disposal. SSbD principles will enhance safety by reducing or eliminating harmful substances. SSbD guidelines should also promote circularity by encouraging recyclability and reuse. This approach aligns with broader goals like the European Green Deal and the Chemicals Strategy for Sustainability.

The potential for biobased packaging in relation to the SSbD principles is quite promising. Biobased packaging materials are derived from renewable resources, which can significantly reduce the carbon footprint compared to traditional fossil-based materials. The Life Cycle Assessment (LCA) methodology can be an important tool in evaluating the environmental impact of packaging throughout its life cycle—from production to disposal. This helps identify areas where improvements can be made to reduce carbon footprint and waste and aligns well with SSbD principles of minimizing environmental impact. Another important aspect of SSbD is to enhance circularity. Plastic materials should be designed for better recyclability and/or compostability, promoting a circular economy. As an example, when a polyamide barrier layer is replaced by a PEF layer in multilayer bottles, the integral PET/PEF bottle can be recycled in the PET recycle stream [109,110], eliminating the risk of polyamide entering the rPET granules and therewith positively improving the quality of the rPET stream. On the other side of the spectrum, there are ample opportunities to change the biodegradability of biobased plastics (e.g., PHA, PLGA) so that they are well adapted to their end-of-life situation (e.g., home composting, marine environment) [26]. This means biobased packaging can be reused, recycled, or safely decomposed, reducing waste and pollution. By using natural and non-toxic materials, biobased packaging can enhance safety for consumers and the environment. This is crucial for SSbD, which prioritizes the elimination of hazardous substances. Another important aspect of packaging is that several minor components can significantly influence the recyclability of plastics. These components include additives (compatibilizers, stabilizers, processing aids, color neutralizers), inks, labels, and coatings, such as barrier coatings. While they enhance the functionality of packaging, they can complicate recycling processes if they are not designed to be easily removable or compatible with recycling streams. Last but not least, caps and closures made from materials different than the main container can hinder recycling. Ensuring they are made from compatible or easily separable materials can improve recyclability. Therefore, it is important that new packaging solutions are based on innovative designs and testing to further improve safety and sustainability. This includes engaging all stakeholders in the value chain and adapting the SSbD framework based on their feedback as well as new data. It is very important that the developed packaging solutions are aligned with existing and upcoming regulations to ensure the solutions meet safety and sustainability standards.

Projects under Horizon Europe, such as the CBE-JU (Circular Biobased Europe Joint Undertaking) PEFerence flagship project, are focusing on scaling up these innovations to meet industrial needs. The European Commission and other bodies are actively supporting the development and implementation of SSbD frameworks, which include biobased solutions. This regulatory backing helps drive adoption and integration into mainstream packaging solutions. Overall, biobased packaging offers a sustainable and safe alternative that aligns perfectly with SSbD principles, making it a key area for future development and innovation.

## 11. Plastic Packaging Solutions Versus Glass, Aluminum, and Paper Packaging

Table 4 outlines widely used fossil-based plastic packaging and its biobased alternatives. Common substitutes like glass, aluminum, and paper are also frequently used. Recyclable biobased plastics such as PEF show strong potential to replace high-performance formats like glass bottles and aluminum cans. This section examines the benefits and drawbacks of glass, aluminum, and paper packaging compared to plastic, focusing on sustainability, functionality, and recyclability.

Glass is fully recyclable and preserves flavor and nutrients due to its impermeability and chemical inertness. It offers design flexibility and durability when handled properly. However, its production is energy-intensive and costly, with most plants relying on fossil fuels. Glass is heavier than plastic, increasing transport costs, and fragile, requiring careful handling and more storage space.

Aluminum provides excellent protection against oxygen, light, and moisture. It is lightweight, corrosion-resistant, and maintains product quality over time. It supports efficient production and is fully recyclable. However, aluminum is more expensive and energy-intensive to produce and contributes to pollution. It often requires fossil-based coatings, may react with acidic foods, and is prone to dents. Its production also has environmental impacts due to mining and waste.

Paper packaging is biodegradable, renewable, and recyclable. It is cost-effective, easy to customize, and popular among environmentally conscious consumers. With added layers, it can protect products effectively. However, paper is sensitive to moisture, lacks strong barrier properties, and is less durable for heavy or fragile items. Its production can lead to deforestation and high energy use.

## 12. Opportunities and Threats of European Regulations and Directives

European regulations and directives are crucial for fostering the growth and success of the biobased plastics industry. Key aspects include standardization and clarity, market confidence, and harmonization. Regulatory frameworks build trust among consumers and investors by ensuring biobased plastics meet specific environmental and safety standards, driving market adoption and investment in bioplastic technologies. European policies, such as the Clean Industrial Deal, the Circular Economy Action Plan, and the forthcoming Circular Economy Act, set ambitious targets for reducing carbon emissions and promoting sustainable materials. These policies encourage the development and use of biobased plastics as part of broader efforts to achieve climate neutrality. The most important policy instruments, regulations, and directives are discussed below.

a.Clean Industrial Deal

The Clean Industrial Deal is a strategic initiative by the European Commission to boost competitiveness and decarbonization across EU industries, including chemicals. It outlines concrete actions to make decarbonization a growth driver, focusing on energy-intensive sectors.

Key elements include the Industrial Decarbonization Accelerator Act, which streamlines permitting and supports modernization and low-carbon product labeling. The Chemicals Industry Package, expected by the end of 2025, highlights the sector’s strategic role and proposes legislation to support its transformation. Circular economy principles are emphasized to reduce dependence on external raw materials and promote recycling, reuse, and sustainable production.

A new State Aid Framework will fast-track support for renewable energy, decarbonization, and clean tech manufacturing. Over €100 billion will be mobilized through strengthened innovation funding, a proposed Industrial Decarbonization Bank, and Horizon Europe calls to drive research and investment. The Carbon Border Adjustment Mechanism (CBAM) will expand to more sectors, offering predictability for the chemical industry.

The Clean Industrial Deal aims to rejuvenate the (bio-)chemical and (bio-)plastics industry by providing the necessary support for decarbonization, modernization, and sustainable practices. By addressing high energy costs, unfair global competition, and complex regulations, the deal seeks to create a more competitive and resilient (bio-)chemical and (bio-)plastic sector in Europe. In addition, several European countries have asked for the establishment of the Critical Chemicals Act, identifying several low-carbon footprint molecules (a.o. lactic acid, FDCA, propanediol, succinic acid) as strategic, ensuring sustainable growth in the sector [111,112].

b.EU policy framework on biobased, biodegradable, and compostable plastics

In 2022, the European Commission adopted a policy framework focused on the sourcing, labelling, and use of biobased, biodegradable, and compostable plastics [113]. This framework was introduced as part of the European Green Deal, Circular Economy Action Plan, and Plastics Strategy, with the goal of contributing to a sustainable plastics economy. It aims to enhance the understanding of biobased plastics by clarifying where these innovative materials can provide environmental benefits, under what conditions, and for which applications. Additionally, the framework ensures that biobased plastics are held to the same strict standards as any other material. It should be noted that the policy framework does not impose any legal obligations.

c.Packaging and Packaging Waste Regulation (EU 2025/40)

The PPWR is a comprehensive framework aimed at reducing packaging waste and promoting sustainability across the European Union. The PPWR entered into force on 11 February 2025, with its general date of application set for 18 months later. It replaces the previous Packaging and Packaging Waste Directive (94/62/EC) and seeks to create a more sustainable and circular economy for packaging in the EU (European Commission 2025 [8]).

The regulation aims to reduce packaging waste by 15% per capita in each member state by 2040, compared to 2018 levels. By 2030, all packaging on the EU market must be recyclable in an economically viable way. For plastic packaging, member states must reach a recycling rate of 55% by 2030. The PPWR promotes reuse and refill systems, requiring companies to offer a portion of their products in reusable or refillable packaging. It also seeks to reduce virgin material use and guide the sector toward climate neutrality by 2050.

Key measures include restrictions on certain single-use plastics, such as pre-packed fruit and vegetables weighing less than 1.5 kg and single-serving condiments. Packaging must be minimized in weight and volume, avoiding unnecessary materials. Targets are set for recycled content: for example, single-use plastic beverage bottles must contain a minimum of 30% and 65% recycled content by 2030 and 2040, respectively. Although no biobased targets are defined yet, the regulation introduces a legal basis for future biobased content requirements.

The PPWR also harmonizes national rules to strengthen the internal market for secondary raw materials, manufacturing, recycling, and reuse, supporting a more sustainable and competitive packaging industry across Europe.

d.Extended Producer Responsibility

Extended Producer Responsibility (EPR) is a policy that holds producers responsible for the end-of-life management of their products, including take-back, recycling, and disposal. Producers often join PROs to manage packaging waste, and they must comply with regulations and report their efforts.

Shifting EPR fees towards eco-modulation encourages sustainable packaging by factoring in the environmental impact and design. This approach incentivizes producers to adopt sustainable packaging solutions. Including recycled and biobased content targets in eco-modulation fees can further boost sustainability efforts by ensuring packaging is both renewable and responsibly managed throughout its life cycle.

EPR systems differ widely across Europe, shaped by national priorities and regulations. Germany, a pioneer since 1991, operates under the Packaging Act (VerpackG), requiring producers to register and join a PRO to manage packaging waste. France’s system, governed by the Circular Economy Law, imposes strict recyclability requirements and eco-modulation fees based on environmental impact. Italy’s CONAI (National Packaging Consortium) oversees the collection and recycling of packaging waste, with producer contributions based on packaging type and volume. The Netherlands has a comprehensive system with mandatory recycling targets and eco-modulation fees, requiring producers to register with the Dutch Packaging Waste Fund and report annually.

e.Implementation in the EU member states

Implementing EU plastics regulations across member states poses several challenges. Although regulations aim for consistency, national interpretations often differ, leading to enforcement discrepancies. The shift from flexible directives to directly applicable regulations is meant to reduce these gaps but requires major national adjustments. Successful implementation depends on robust recycling and waste management infrastructure, which many countries still need to upgrade. This demands significant investment and can be financially burdensome for some member states.

Compliance with new regulations often involves additional costs from compliance, such as adopting new technologies or packaging materials, which can be especially difficult for SMEs. The market for recycled and biobased materials remains limited and expensive. Clear labeling among member states is essential to guide consumers, but enforcement is resource-intensive and uneven. Strong monitoring, penalties, and incentives are needed to ensure compliance.

Addressing these issues requires coordinated EU and national efforts, including infrastructure, investment in business support, public awareness, and consistent enforcement.

## 13. Scaling the Biobased Industry in Europe

Currently, there is no level playing field in Europe for biobased plastics compared to fossil-based plastics. Several factors contribute to this imbalance. Firstly, there is no comprehensive EU law specifically for biobased plastics. Consistent and supportive policies are crucial. The European Commission’s Clean Industrial Deal, for example, aims to accelerate reindustrialization and decarbonization, which includes promoting biobased industries, but it needs further development [114]. Fossil-based products are still subsidized, and very few of the societal and environmental costs of fossil-based plastics are allocated to plastic packaging solutions.

One of the major challenges for biobased plastics is their cost relative to performance. Biobased plastics are generally more expensive to produce due to the cost of raw materials and the production processes used. Therefore, scaling the biobased plastics industry in Europe is important to reduce costs. However, several factors hamper scaling. Scaling of (bio-)chemical processes is inherently very capital-intensive, with current Capex costs for most biobased polymers ranging from €5000 to €40,000 per ton of product. To facilitate scaling, strong market demand needs to be created and fostered through public procurement policies, low-carbon product labels, and clear standards for sustainable products to drive market growth [115].

There is often a chicken-and-egg dilemma when it comes to scaling up production volumes of biobased plastics without a guaranteed market demand. To address this, Europe should actively support the market uptake of biobased plastics, for example, by including biobased plastics in the target setting of the PPWR, ELV (End-of-Life Vehicle) Regulation, and eco-modulation design criteria. Furthermore, biobased building blocks—particularly drop-in molecules—are highly sensitive to fluctuations in oil prices, which can undermine their competitiveness and investment stability.

Leveraging both public and private funding is essential. Initiatives like the CBE-JU provide significant funding to innovative projects, helping to partly de-risk investments and attract private capital. Other tools, such as the Innovation Fund and the European Investment Bank (EIB), should also be open to funding sustainable biobased plastic initiatives, including those based on (side streams from) first-generation crops such as starch and sugar, as long as the cascading principle and sustainability criteria are followed. Energy and raw material costs should be competitive at a global level, or their effects should be mitigated through measures such as CBAM.

## 14. Conclusions and Outlook Based on a SWOT Analysis

Below in Table 5, a SWOT (Strengths, Weaknesses, Opportunities, Threats) analysis is prepared, summarizing the major findings discussed in this paper. The strengths highlight the potential of biobased plastics to contribute to a more sustainable future, while the weaknesses highlight the challenges that need to be addressed for biobased plastics to become a viable and sustainable bulk alternative to conventional plastics. The opportunities emphasize the potential for biobased plastics to contribute to a more sustainable future. However, the threats stress the challenges that need to be addressed for biobased plastics to become a viable and sustainable alternative to conventional plastics at scale.

The outlook for biobased plastics is promising, driven by their numerous strengths and opportunities. Biobased plastics, derived from renewable resources, offer unique functionality, a reduced carbon footprint, recyclability, and, when desired, biodegradability, addressing critical environmental concerns. Their non-toxic nature and energy-efficient production further enhance their appeal. The versatility of biobased plastics allows for wide-ranging packaging applications, from high-performance barrier layers in multilayer bottles all the way to thermoforms and plastic bags. Innovation potential and government support are key drivers, with ongoing research leading to new materials and applications. The growing consumer demand for sustainable products and corporate sustainability goals are propelling market growth. Regulatory support and incentives, such as inclusion in the Packaging and Packaging Waste Regulation and Extended Producer Responsibility schemes, can create a favorable environment for biobased plastics.

However, challenges remain. Higher production costs and limited end-of-life infrastructure hinder widespread adoption. Land use concerns and energy-intensive production can reduce environmental benefits. Consumer confusion about disposal and perceived contamination issues complicates waste management. Regulatory inconsistencies and market penetration difficulties pose additional barriers. Despite these threats, the biobased plastics market is expected to grow significantly, offering substantial business opportunities. Improved infrastructure, technological advancements, and continued research and development will be crucial in overcoming these challenges and realizing the full potential of biobased plastics in promoting a circular economy and reducing plastic pollution.

## Figures and Tables

**Figure 1 polymers-17-02217-f001:**
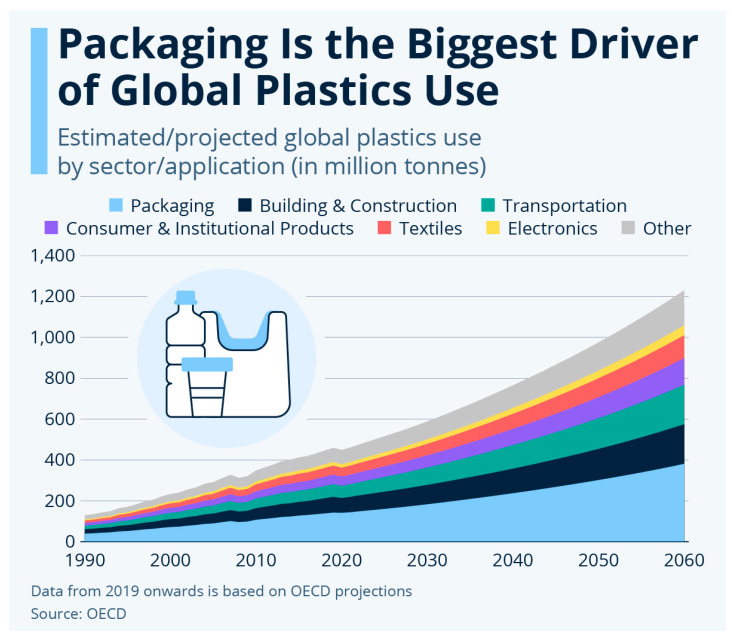
Packaging is currently the largest application of plastics and is projected to remain the dominant use through 2060 [6].

**Figure 3 polymers-17-02217-f003:**
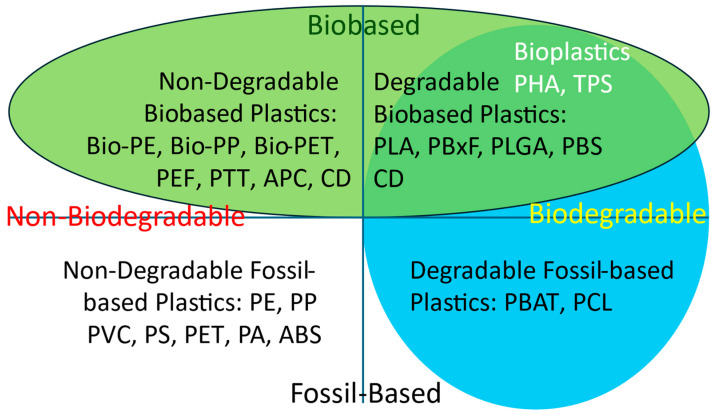
Polymer classification along four axes (non-biodegradable and biodegradable (*X*-axis), biobased and fossil-based (*Y*-axis). This figure is an adaptation of Gruter [2], which was based on the system developed by European Bioplastics (www.european-bioplastics.org/ (accessed on 25 May 2025)). Abbreviations are explained in Section 5 and Section 6, except for PCL = polycaprolactone [26].

**Figure 4 polymers-17-02217-f004:**
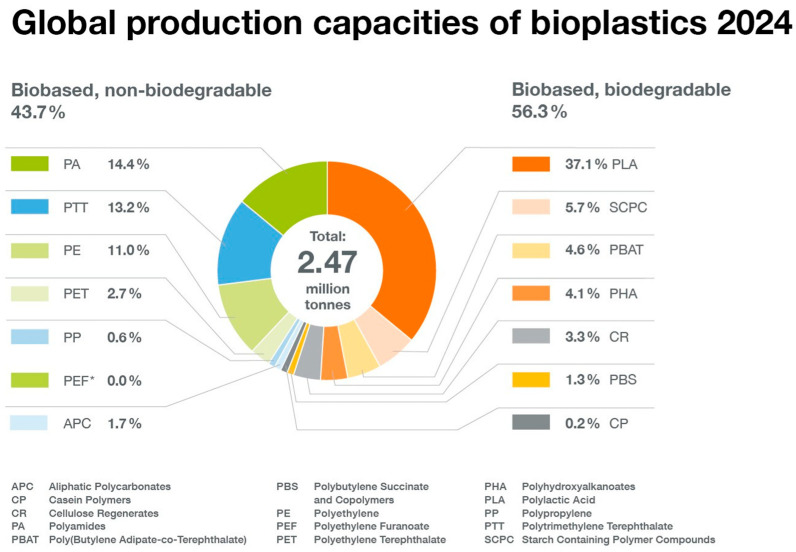
The global production capacities for both biodegradable and non-biodegradable biobased plastics. * The PEF is available at a commercial scale as of 2024. Source: European Bioplastics, nova Institute [36].

**Figure 5 polymers-17-02217-f005:**
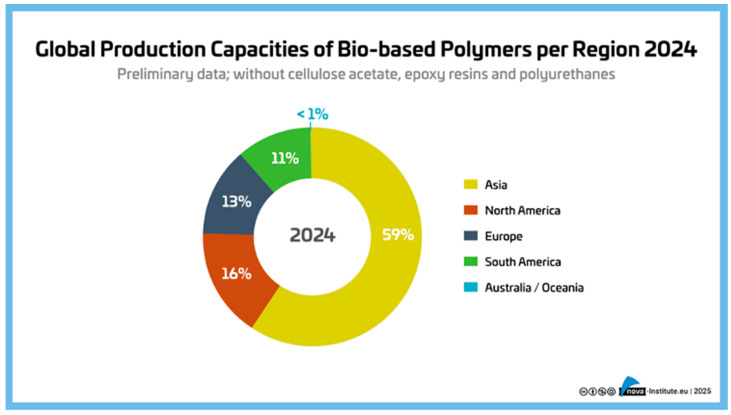
The global production capacities of biobased polymers per region in 2024 [36].

**Table 2 polymers-17-02217-t002:** Comparison of the pros and cons of drop-in polymers versus novel polymers for a wide range of criteria. The more ++, the better the drop-in or novel polymer is adapted to that situation.

Property	Drop-Ins (e.g., bio-PE, bio-PP, bio-PET)	Novel Polymers (e.g., PEF, PLA, PHA)
Time to market	+++ *	+/− *
Improved properties	0	−/+++
CAPEX	−−−−	−−
Own market space	−−−	+++
Prices of fossil incumbents	−−−−	++
Regulatory (Reach/EFSA)	++	−−−
Legislation (a.o. PPWR)	−	−−−
Value chain	++	−−
Circularity	+++	−−/+++
Recycling infrastructure	++	−−−
Production cost from glucose	−−	0

* 0 = Comparable to the fossil-based alternative; − = Inferior to the fossil-based alternative; more minus signs indicate greater inferiority; + = Superior to the fossil-based alternative; more plus signs indicate greater superiority; +/− = Performance varies depending on the specific polymer.

**Table 4 polymers-17-02217-t004:** Overview of the most important fossil-based plastic packaging solution and the potential biobased alternatives. Bio-PET and bio-PTT are currently only partially biobased (MEG or propanediol part) [26,27,65,66,67,72,73,74,75,76,77].

Packaging Solution	Fossil-Based Plastic	Biobased Plastic
Carbonated soft drink bottles (CSDs), juice bottles, carbonated water bottles	PET, PA in multilayer	PEF, bio-PET,
Water and milk bottles	PET, HDPE	PEF, Bio-PET, Bio-PE
Shampoo and detergent bottles	HDPE, PET, LDPP, PVC	PEF, Bio-PE, Bio-PP, PLA
Food containers	PC, HDPE, HDPP, PET	PLA, Bio-PE, Bio-PP, Bio-PTT
Cosmetic containers	PC, ABS, PBAT, PET, HDPP	PEF, PBxF (a.o. PBAF), PLA, PHA
Microwavable containers	PP, LDPE	Bio-PP, bio-PE
Thermoforms (a.o. trays, clamshells, blister packs)	PET, PVC, PS, PP, HDPE	PEF, Bio-PET, PLA, PHA, TPS, Bio-PP, bio-PE
Agricultural packaging	PBAT, HDPE, LDPE	PBxF, PHA, TPS. Bio-PE
Protective packaging (a.o. electronics, pharmaceuticals)	PA, PC, ABS, PVC, PS, LDPP, LDPE	PEF, PLA, TPS (foamed)
Multi-layer flexible packaging (a.o. pouches, films)	PA, PE, PP, PET	PEF, Bio-PE, Bio-PET, CD
Barrier films and coatings	PA, PET, PP, PE	PEF, PHA, PLGA
Stretch, shrink, and cling films	PVC, LDPE	TPS, PLA, PHA
Liners (a.o. cereal boxes)	LDPE, HDPE, PET	Bio-PE, PLA, PHA
Shopping bags	LDPE, HDPE, LDPP,	TPS, PLA, Bio-PE
Bottle sleeving, caps, and closures	HDPP, HDPE, PVC, PET	PLA, PHA

**Table 5 polymers-17-02217-t005:** A Strengths, Weaknesses, Opportunities, and Threats (SWOT) analysis of the growth potential of biobased plastics.

Strength	Weaknesses
**Renewable Resources**: Biobased plastics are made from renewable biological sources like starch and sugar, as well as waste streams, reducing dependence on fossil fuels. **Reduced Carbon Footprint**: The production of biobased plastics generally emits many fewer greenhouse gases compared to conventional plastics. **Biodegradability**: Several biobased plastics are designed to biodegrade, helping to reduce plastic pollution and waste management issues for those applications where recycling is not an option. **Non-Toxic**: Biobased plastics typically do not contain harmful chemicals, making them safer for both human health and the environment. **Versatility**: Biobased plastics can be used in a wide range of packaging applications. **Support for a Circular Economy**: Biobased plastics can be integrated into circular economy models, promoting recycling and reuse. Starting with biobased carbon creates a true circular economy. **Market Demand**: Increasing consumer awareness and demand for sustainable products drive the growth of the biobased plastics market.	**Higher Production Costs**: Biobased plastics are generally more expensive to produce than conventional plastics due to the cost of raw materials, maturity, and scale of processing technologies. **Limited End-of-Life Infrastructure**: Many regions lack the necessary infrastructure to properly recycle or compost biobased plastics, leading to improper disposal. **Biodegradation Conditions**: Many biodegradable plastics require specific (industrial) composting conditions to degrade, which are not always available. **Market Penetration**: The biobased plastics market is still relatively small, making it challenging to achieve economies of scale. **Consumer Awareness**: There is often confusion among consumers about the difference between biobased and biodegradable plastics, which also influences the proper disposal of biobased plastics. **Environmental Impact**: The environmental benefits of biobased plastics can be negated if they are not disposed of correctly, as they can still contribute to pollution. **Government Support:** Although some governments offer incentives and support for the research, development, and use of biobased plastics, more needs to be done to facilitate industry growth.
**Opportunities**	**Threats**
**Government Incentives**: Subsidies, tax breaks, and other incentives encourage investment in biobased plastics research and development. **Regulatory Support**: Policies aimed at reducing plastic pollution, such as plastic bag bans and EPR schemes, can create a strong market for biobased plastics, with inclusion into the PPWR generating a strong boost. **Technological Advancements**: Innovations in bioplastic production and processing techniques can improve performance and reduce costs. **Innovation Potential**: The biobased plastics industry is rapidly evolving, with ongoing research leading to new materials and applications. **Advancement of Human Capital:** Strengthening collaboration with educational institutions to align curricula with the evolving needs of the biobased industry enables the development of a skilled and adaptable workforce that supports innovation and long-term growth. **Expansion into New Markets**: Biobased plastics can penetrate various industries, including automotive, electronics, and medical sectors. **Improved Infrastructure**: The development of robust collection, recycling, and composting infrastructure can enhance the viability of biobased plastics. **Corporate Sustainability Goals**: Companies are increasingly adopting biobased plastics to meet their sustainability targets and reduce their environmental footprint. **Global Market Growth**: The global biobased plastics market is expected to grow significantly, offering substantial business opportunities. **Security of Supply**: A healthy European biobased plastics industry can have positive societal impacts by moving a big part of the needed feedstock production and associated logistics to local farmers and entrepreneurs.	**Abundant and Cheap Oil:** Biobased plastics are already often more expensive to produce than conventional plastics; flooding the market with cheap oil is a serious threat in the current political context. **Raw Material Supply**: The availability and cost of raw materials for biobased plastics can be affected by agricultural conditions and market fluctuations. **Land Use and Food Security**: The cultivation of crops for biobased plastics uses land, which can raise concerns in the public’s opinion about food security. **European Investment Bank Policy**: The current exclusion of first-generation feedstocks as eligible biomass for EIB funding severely hinders the scaling of the bioeconomy in the EU. **Consumer Confusion**: There is often confusion among consumers about the difference between biobased and biodegradable plastics. This results in improper disposal of biobased plastics, leading to contamination of recycling streams. **Perceived Contamination Issues**: Novel plastics, including biobased plastics, can increase the complexity of recycling streams if they are not properly sorted, complicating waste management. **Environmental Impact**: If not disposed of correctly, biobased plastics can still contribute to pollution, including littering and the formation of microplastics. **Limited Industrial Composting Facilities**: The lack of industrial composting facilities in some EU countries can hinder proper disposal. **Regulatory Challenges**: Inconsistent regulations and standards across different regions can create barriers to the adoption and growth of biobased plastics. In addition, new polymers that want to enter the food contact market are scrutinized according to the newest (more stringent) regulations. **Market Penetration**: The biobased plastics market is still relatively small, making it challenging to achieve economies of scale and widespread adoption. **Public Perception**: Skepticism about the true environmental benefits of biobased plastics can affect consumer acceptance and demand. **Misleading Claims:** Wrongly labelled products can mislead consumers and undermine trust.

## Data Availability

No new data were created or analyzed in this study.

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
