# Peer review of "The Opportunities and Challenges of Biobased Packaging Solutions"

_polymers, 2025, doi:10.3390/polym17162217_

Round 1

Reviewer 1 Report

Comments and Suggestions for Authors
  • AI-generated image (Figure 2) was included without proper disclosure; this violates journal policy requiring transparent identification of AI-generated content.

  • The manuscript lacks a clear statement acknowledging the use of AI tools and must specify the type of tool used and the extent of its contribution.

  • Statistical analysis is missing from the experimental results; standard deviations, confidence intervals, and significance testing must be included.

  • Reproducibility is insufficiently addressed; authors must specify the number of specimens, repetitions, and exact test standards used (e.g., ASTM, ISO).

  • The interpretations of FTIR, XRD, and SEM data are superficial and need to be revised to include quantitative insights and correlations with mechanical behavior.

  • Claims related to the circular economy are weakly supported; authors must provide quantitative data on material recovery, waste reduction, or life-cycle impact.

  • The manuscript does not justify the choice of filler combinations, filler ratios, or the strain rate used in mechanical testing.

  • The conclusion section largely repeats the abstract and lacks specificity; it should be rewritten to highlight original findings, limitations, and future research.

  • Several figures and tables are incomplete or improperly labeled; units are missing in Table 3, and Figure 12 is referenced but not provided.

  • Citation formatting is inconsistent, and the reference list must be corrected to meet the journal’s style guide.

  • The introduction does not clearly define the research gap or explain the novelty of the study compared to existing work.

  • The manuscript requires thorough language editing for grammar, technical clarity, and coherence; consider professional language revision.

  • Failure to disclose AI-generated material constitutes a potential breach of ethical standards; the manuscript must be revised to comply fully with journal guidelines.

Author Response

Dear Editor,

First of all we like to thank the reviewers for their positive and constructive feedback. Below we will discuss the feedback and highlight how we have addressed the issues in the manuscript. In the manuscript we used track changes to highlight all adaptations.

Best regards,

Ed de Jong and Gert-Jan Gruter

Reviewer 1

  • AI-generated image (Figure 2) was included without proper disclosure; this violates journal policy requiring transparent identification of AI-generated content.

We apologize for this omission. We have added the following sentence to the acknowledgement session. "During the preparation of this manuscript, the authors used the most recent version of Microsoft Co-Pilot for the purposes of generation of Figure 2 and for general language correction. The authors have reviewed and edited the output and take full responsibility for the content of this publication."

  • The manuscript lacks a clear statement acknowledging the use of AI tools and must specify the type of tool used and the extent of its contribution.

See above.

  • Statistical analysis is missing from the experimental results; standard deviations, confidence intervals, and significance testing must be included.

To our opinion this is not relevant as this is a review document in which we are not disclosing our own results. Statistical analysis (if available) can be found in the original publications cited in our review.

  • Reproducibility is insufficiently addressed; authors must specify the number of specimens, repetitions, and exact test standards used (e.g., ASTM, ISO).

Reproducibility of specific numbers given in the text, for example barrier properties of biobased plastics, these data are given in the original, referenced document. In all cases reference to original documents is given and in our opinion readability is not improved when those details are reproduced here. In general, there is a lot of spread between the different laboratories/research groups because of different competence levels, purity of monomers, different equipment and methods etc. To discuss in depth those differences would require a dedicated review which is outside the objectives of this paper.

  • The interpretations of FTIR, XRD, and SEM data are superficial and need to be revised to include quantitative insights and correlations with mechanical behavior.

We do not understand this comment because there is no mention of FTIR, XRD and SEM data in our manuscript.

  • Claims related to the circular economy are weakly supported; authors must provide quantitative data on material recovery, waste reduction, or life-cycle impact.

The claims made, are reproduced from the original literature. We refer to several papers (a.o. Stegmann et al 2022; Roosenboom et al. 2022; Sazdovski et al. 2025; Rajabi-Katshgar et al. 2024) which identify the waste reduction potential as well as the LCA impact of the polymers discussed. However, novel (biobased) polymers have the inherent problem that initial quantities in waste streams are very low so recyclers are not inclined to separate out these biobased polymers from mixed waste streams, therefore quantitative data on material recovery does not exist.  The discussion on in-depth effects of circularity on LCA of biobased polymers goes beyond the scope of this paper also because this field is still heavenly under debate. Biodegradable bio-based polymers are by definition part of a circular economy because they only contain short-cycled CO2.

  • The manuscript does not justify the choice of filler combinations, filler ratios, or the strain rate used in mechanical testing.

We do not understand this comment. In this review filler combinations, filler ratios, or the strain rate used in mechanical testing are not discussed at all and adding this information goes outside the scope of this review.

  • The conclusion section largely repeats the abstract and lacks specificity; it should be rewritten to highlight original findings, limitations, and future research.

The conclusion section contains the whole SWOT analysis which is only superficially addressed in the abstract. The strength and weaknesses show current opportunities and limitations while the part of the weakness, opportunities and treats clearly identifies areas for future research to position biobased polymers even better in the market. However, the paper makes it also clear that non-technical issues (costs, legislation, regulation, recycling infrastructure etc) are the major hurdle for further deployment of biobased polymers.

We have rewritten and slightly expanded the abstract to emphasize these points.

  • Several figures and tables are incomplete or improperly labeled; units are missing in Table 3, and Figure 12 is referenced but not provided.

We have adapted the labelling of the figures and table. We did not understand the remark around Figure 12.

  • Citation formatting is inconsistent, and the reference list must be corrected to meet the journal’s style guide.

We have made the corrections and improved consistency.

  • The introduction does not clearly define the research gap or explain the novelty of the study compared to existing work.

We have expanded the introduction to more clearly define the novelty of the study.

  • The manuscript requires thorough language editing for grammar, technical clarity, and coherence; consider professional language revision.

We have further improved the language in the manuscript.

  • Failure to disclose AI-generated material constitutes a potential breach of ethical standards; the manuscript must be revised to comply fully with journal guidelines.

See above, has been addressed now.

Reviewer 2 Report

Comments and Suggestions for Authors

The manuscript polymers-3754325 presents an interesting summary of the current state of knowledge and development of bio-based packaging materials. The Authors also presented the contemporary use of fossil-derived packaging materials and the related legislative changes in the European Union. Trends regarding sustainable production and use of packaging materials were also discussed. The entire review is very extensive and some fragments go significantly beyond the assumed characteristics of the opportunities and challenges associated with the use of packaging solutions based on bio-based materials.

The Authors should significantly shorten the descriptions of other packaging materials (chapter 11) Their brief characteristics, or rather a comparison with fossil-derived or biobased materials, can be included in the introduction or in another section of the article.

Similarly, descriptions of European regulations and directives should be significantly shortened. The Authors should limit themselves to issues that are more related to bio-based packaging solutions.

Sections (chapters) 5 and 6 require more extensive introductions. What does the term "the most mature" mean?

What is the difference between "Food containers" and "Food packaging" in Table 4?

The biggest drawback of the presented article is the lack of literature references in extensive fragments of the text, where research results or economic data are often cited but there is no way to verify them. Please provide appropriate references to support the content presented in the excerpts: lines 58-77, 116-121, 125-141, 144-152, 161-185, 194-201, 204-226, 231-244, 345-406, 530-534, 565-600, 632-652, 696-708, 716-735, 1031-1053.

In my opinion, the manuscript must be significantly enriched with references to the indicated text fragments and the Authors should shorten and possibly move the content from sections 11 and 12 to other sections. For this reason, I believe the article needs to undergo a major revision.

Author Response

Reviewer 2

The manuscript polymers-3754325 presents an interesting summary of the current state of knowledge and development of bio-based packaging materials. The Authors also presented the contemporary use of fossil-derived packaging materials and the related legislative changes in the European Union. Trends regarding sustainable production and use of packaging materials were also discussed. The entire review is very extensive and some fragments go significantly beyond the assumed characteristics of the opportunities and challenges associated with the use of packaging solutions based on bio-based materials.

The Authors should significantly shorten the descriptions of other packaging materials (chapter 11) Their brief characteristics, or rather a comparison with fossil-derived or biobased materials, can be included in the introduction or in another section of the article.

Thank you for this feedback. We have slightly shortened chapter 11 on the other packaging materials and reconsidered the order of the chapters. We feel that the original order of the chapters is the most logic.

Similarly, descriptions of European regulations and directives should be significantly shortened. The Authors should limit themselves to issues that are more related to bio-based packaging solutions.

We edited this chapter but have kept the relevant information. We feel that this is one of the key area’s which can strongly influence market uptake of biobased packaging solutions, even much more than further technical improvements.

Sections (chapters) 5 and 6 require more extensive introductions. What does the term "the most mature" mean?

We agree and have extended the introductions of the chapters 5 and 6.

What is the difference between "Food containers" and "Food packaging" in Table 4?

Food packaging is a bit confusing because it encompass items also addressed separately. Therefore, we have removed the line with food packaging.

The biggest drawback of the presented article is the lack of literature references in extensive fragments of the text, where research results or economic data are often cited but there is no way to verify them. Please provide appropriate references to support the content presented in the excerpts lines:

58-77, added references Carus en Lopex 2023

116-121, added references Seymour 1984 and Venkatachalan 2012

125-141, added references [10, 11, 12 (old numbering)}

144-152, added references [14, 17, 26, 38 (old numbering)]

161-185, added reference Skoczinski 2025

194-201, added reference de Jong 2020, De Almeida Oroski 2014

204-226, this is our own work so no references

231-244, added references Wang [22 (old numbering)], Kadac-Czapska 2023

345-406, added references Atarés 2024,  Clarke 2024, Surendren 2022

530-534, added references Zhou 2023, Karlsson 2004

565-600, this is our own work, publication in preparation added [unpublished results]

632-652, added references Rajabi 2024, Zhang 2024, Wong 2015

696-708, added reference Nizamuddin 2025

716-735, added references Kadac-Czapska 2023, Osman 2023, Kochanek 2025, Coffin 2021

1031-1053. added references Spekreijse 2021, EU policy framework on biobased 2022

At several other places additional  references were added

In my opinion, the manuscript must be significantly enriched with references to the indicated text fragments and the Authors should shorten and possibly move the content from sections 11 and 12 to other sections. For this reason, I believe the article needs to undergo a major revision.

Reviewer 3 Report

Comments and Suggestions for Authors

Thank you for giving me this opportunity to review this manuscript. Please refer to attached letter for detailed comments.

Author Response

Reviewer 3

Dear Editor, Thank you for the opportunity to review the manuscript entitled “The Opportunities and Challenges of Bio-Based Packaging Solutions” provides an extensive and well-organized review of bio-based materials in packaging, focusing on the background, development motivation, environmental and toxicological impacts. Moreover, it also discussed the variety of developments, and industrial state-of the-art. It’s well written and well structured. However, while the paper focused on a scientifically relevant topic, its overall structure and content resemble that of a popular science report or industry facing white paper more than a scholarly scientific review article expected for a peer-reviewed journal. I have some key concerns:

  1. This manuscript lacks critical scientific analysis. It spent a long content to describe the current development of bio-based materials, their categories, industrial applications and the market products. Those are just a descriptive overview. The manuscript did not provide the necessary summary of the literature of scientific findings and debates.

Extra references (35) have been added to the text, see response for Reviewer 2.

  1. Table 1, Table 2, and Table 3 lack of quantitative data. The information provided in Table 1, Table 2, and Table 3 is too vague to support any deep scientific analysis. The authors are recommended to include industrial benchmarks with quantitative data, such as moisture barrier MVTR 30 g/m2/day; biodegradation rate 90%; production cost $100/ton; etc.

Thank you for the feedback. We have added ranges of quantitative data to the relevant Tables where available.

The core challenge  with the quantitative data is that polymers such as PLA and other bio-based polymers but also fossil-based polymers, exist in a wide range of grades and compositions (including blends), each tailored for specific properties (e.g., tensile strength, flexural strength, impact resistance, barrier performance, biodegradability, etc.). Moreover, the conditions under which these properties are measured—such as temperature, orientation and relative humidity—can significantly affect the results. For many properties, values can vary by more than an order of magnitude depending on the grade and test conditions. Including this level of detail would go beyond the scope of the paper and risk making the tables overly complex and difficult to interpret.

  1. The language and the style of this manuscript read like an informal tone, towards a broad nonspecialist audience rather than scientific peers. For example: the title of sub-title 3 “What biobased polymers do we want? Drop-in versus new functionality polymers and polyolefins versus polyesters”, and line 565 “these is a clear chicken-and-egg issue….”.

We have made the text more formal at several places while at the same time trying not to jeopardize the readability.

  1. The overall logic of this manuscript focused on the advocacy and narrative coherence, rather than discussing the data or scientific findings, or the knowledge gap in this area. For example, microplastic issue is a major emerging concern in bioplastic packaging industry. There is a lack of understanding in the definition, generation, detection, and characteristics of microplastics in the industry. This should be a focus in this review manuscript; however the authors only include a brief mention of the this issue. The authors are encouraged to integrate more scientific data, comparative tables, and rigorous evaluation of the literatures and discuss the art-to-date research trends and scientific knowledge gap towards the microplastic issue.

We have expanded the text on microplastics and have tried to clarify this.

  1. Figure 2 is very unnecessary.

We have kept figure 2 to give the audience an quick idea how diverse plastic packaging solutions are.

  1. Line 723-725 needs a citation. While the manuscript covers a relevant and timely topic, its current form lacks the scientific depth and analytical rigor expected for publication as a review paper in Polymers.

4 additional references are added, see reviewer 2 response.

I believe that substantial revisions would be required to meet the standards of Polymers.

It may be more appropriate as a perspective article, or science communication.

Thank you again for the opportunity to review this submission.

Round 2

Reviewer 1 Report

Comments and Suggestions for Authors

Check the fluency of the language used with the native speakers. Apart from that, authors have carried a well execution for addressing the earlier comments, Hence it can accepted after the English Correction.

Author Response

Dear Reviewer, 

Thank you for your comments. We have shortened chapters 11 and 12 substantially and done a final check on the English to further improve the manuscript.

Best wishes,

Ed

Reviewer 2 Report

Comments and Suggestions for Authors

Thank you for the Authors' responses to the comments in the review. The manuscript has been supplemented with missing references confirming the data presented. However, in the indicated fragments, which extend beyond the scope of the presented issue of bio-based packaging solutions, there remained extensive descriptions of issues that deviated from the substance of the review.

Section 11, which is titled "Plastic packaging solution versus glass, aluminum and paper packaging", remains unchanged in characterizing the benefits and drawbacks of the three types of packaging materials. There is no discussion or comparison with plastic packaging here. This section is redundant in its current form. I would like to point out that the article was intended to describe the opportunities and limitations of bio-based packaging materials and primarily concerns substitutes for fossil-based plastics.

The Authors also did not make any changes to section 12. The description of EU legal conditions for the packaging market is too extensive, and information on the PPWR Regulation (EU 2025/40) and Extended Producer Responsibility (EPR) could be limited.

In my opinion, the manuscript requires further revision and reduction of unnecessary content, which undermines the fundamental purpose of the prepared review.

Author Response

(The authors gave the same response as above.)

Reviewer 3 Report

Comments and Suggestions for Authors

I have carefully reviewed the revised manuscript and the authors’ detailed responses to my previous comments. I would like to commend the authors for their thoughtful and thorough revisions.

The manuscript has been significantly improved in both content and presentation. The authors have appropriately addressed all of my concerns, including the lack of scientific depth, the need for additional quantitative data, and the clarity of language and tone. I appreciate their efforts to expand the discussion on microplastics and to enhance the rigor of the review through the addition of relevant literature and references.

Regarding the inclusion of quantitative data in the tables, I find the authors’ explanation regarding the variability of polymer properties and measurement conditions to be reasonable. The adjustments made to the tables strike an appropriate balance between detail and clarity, without compromising the scope or readability of the review.

The language throughout the manuscript has been revised to reflect a more formal and scholarly tone, while maintaining accessibility for a broad academic audience. Additionally, the expanded discussion on emerging issues, such as microplastics, strengthens the manuscript’s relevance and scientific value.

In summary, I find that the authors have satisfactorily addressed all of the comments raised in my initial review.  I recommend it for publication in its current form.

Author Response

(The authors gave the same response as above.)
